# Parental Bonding and Relationships with Friends and Siblings in Adolescents with Depression

**DOI:** 10.3390/ijerph19116530

**Published:** 2022-05-27

**Authors:** Sarah Christine Fahs, Randi Ulberg, Hanne-Sofie Johnsen Dahl, Per Andreas Høglend

**Affiliations:** 1Division of Mental Health and Addiction, Institute of Clinical Medicine, Blindern, University of Oslo, P.O. Box 1171, 0318 Oslo, Norway; randi.ulberg@medisin.uio.no (R.U.); p.a.hoglend@medisin.uio.no (P.A.H.); 2Research Unit, Division of Mental Health, Vestfold Hospital Trust, P.O. Box 2169, 3125 Tønsberg, Norway; h.s.j.dahl@psykologi.uio.no; 3Department of Psychiatry, Diakonhjemmet Hospital, Vinderen, P.O. Box 85, 0319 Oslo, Norway; 4Department of Psychology, Faculty of Social Sciences, Blindern, University of Oslo, P.O. Box 1094, 0317 Oslo, Norway

**Keywords:** parental bonding, adolescent relationships, friendships, sibling relationships

## Abstract

According to attachment theory, the quality of the early child-parent bond determines the child’s interpersonal relationships later in life. Utilising data from The First Experimental Study of Transference Work-In Teenagers (FEST-IT), the current paper investigated the connection between the self-reported quality of bonding with mother and father and the self-reported importance of relationships with friends and siblings in adolescents with depression. The scales employed were the Parental Bonding Instrument (PBI) and the Adolescent Relationship scale (ARS). A Pearson’s correlation tested the relationship between the reported levels of maternal and paternal care and control, and the reported importance of friendship and relationship with siblings. Results revealed a statistically significant negative correlation between high levels of maternal control and importance of friendship, and a statistically significant positive correlation between high levels of paternal care and importance of relationships with siblings. The results are in line with Bowlby’s theory of attachment.

## 1. Introduction

Depression is among the most common psychiatric disorders during adolescence [1]. Adolescence is a risk period for the onset of depression, and the prevalence of depression rises from <1% [2] in childhood to 4% [3] in adolescence. Depression in adolescence has high recurrence rates and poor functional outcomes [4], and increases the risk of self-harm, suicide, physical illness, substance misuse, and interpersonal problems later in life [5].

Studies suggest that the prevalence of depression in adolescence has increased during recent years [6]. In Norway, symptoms of depression in adolescents, particularly in girls, have increased between 2010 and 2019 [7,8]. 65% of high school children described themselves as more depressed during the COVID-19 pandemic than at the start of the pandemic [8]. Early relationships, loss, and relational trauma are thought to have great impact on the risk of developing depression later in life, and negatively affect future relationships [9]. Understanding the relationships of adolescents with depression is therefore important.

The diagnostic criteria used to define depression in adolescence are similar to those used in adults. The diagnosis of Major Depressive Disorder (MDD) is the same in DSM IV [10] and DSM V [11] in young people and depends on symptoms such as sadness, irritability, loss of interest, or loss of pleasure. In contrast to adults, and to avoid overdiagnosis of bipolar disorder in children and adolescents, irritability is defined as a significant symptom of depression in young people [12]. Other symptoms of depression include appetite and/or sleep disturbance, loss of energy, low self-esteem, reduced concentration, social withdrawal, and a sense of hopelessness. In severe cases, there may also be a strong sense of guilt and suicidal ideation.

Attachment theory offers a valuable conceptual model for understanding the connection between interpersonal relationships and depression in adolescents. According to Bowlby, early interactional patterns between the infant and parent creates an attachment bond that is essential to a child’s normal development [13]. Through behaviours such as sucking, cuddling, looking, smiling and crying infants shape the behaviour of their attachment figure, who in turn responds sensitively and appropriately to the infant’s changing needs. It is through this transactional pattern between the infant and caretaker that the attachment bond is created.

According to Bowlby, the attachment bond created in infancy is internalised, and shapes the child’s internal working model of other and of self [13]. The internal working model of other concerns the child’s image of the attachment figure, and whether or not he or she provides stability and security, and responds to calls for support and protection. The internal working model of self concerns the child’s image of itself, and whether or not it judges itself as loveable and worthy of its attachment figure’s care. Once the internal working model of other and self have been shaped, they serve as a prototype for future relationships.

A securely attached individual will have a sense of self-worth and an expectation that others are generally accepting and responsive [14]. Insecure attachment bonds, on the other hand, have been postulated to play a significant role in the development of depressive cognition [15]. Children with insecure attachment bonds may interpret later loss or disappointment as personal failure and be pessimistic regarding their restoration abilities, which disposes towards hopelessness and clinical depression [16]. The hopeless withdrawal that is often characteristic of depression affects the young person’s ability to engage in relationships with his or her peers and create meaningful friendships [9].

Adolescence is a developmental period in which the young person consolidates his or her own independent identity [9]. During this process, parents continue to provide a secure base from which adolescents can explore [17]. However, the focus on parents decreases, as the adolescent learns to depend on a greater network of attachment figures [18]. Research shows that the character of both friendships and sibling relationships change during the adolescent’s search for an independent identity. Particularly friendships take on a unique significance, and friends become providers of companionship, social and emotional support, and intimate self-disclosure and reflection [19,20,21]. At the same time, adolescents spend less time with their siblings, and the sibling relationship becomes less intense and more egalitarian than during childhood [22].

Studies have used a variety of measures to assess the parent-child relationship. Among many functions that determine parenting styles, Parker et al. [23] identified two specific characteristics of parenting styles: care and control. Care is the level of affection, warmth and emotional closeness exhibited by the parent, contrasting with indifference and rejection. Control is the level of overprotection and intrusion showed by the parent, contrasting with allowing independence and autonomy. Along the dimensions of care and control Parker et al. proposed four parenting styles: Optimal Bonding (high care and low control), Affectionate Constraint (high care and high control), Neglectful Parenting (low care and low overprotection) and Affectionless Control (low care and high overprotection) [24].

Parker et al. developed a self-report form to complement his theory [23]. The form is called the Parental Bonding Instrument (PBI), and retrospectively measures the levels of care and control an individual received in childhood. Literature has found that correlations between care and control levels, as measured by the PBI, are linked to psychopathological disorders in adults, such as depression [25,26,27], anxiety disorders [28,29], obsessive-compulsive disorders [30] and schizophrenia [31,32]. Particularly low levels of care have been identified as a risk factor to lifetime depression [33]. Recently, Raffagnato et al. found that adolescents with psychopathology experience parental bonding characterized by lower care and higher overprotection, especially in girls [34].

Furthermore, connections have been made between levels of parental care and control during childhood and the quality of attachment styles in adult relations [35,36,37]. Few studies have researched the process through which the parent-child relationship shapes relationships during adolescence. Among notable findings, Boling et al. [38] and Lieberman et al. [39,40] have found connections between parents’ attachment styles and the quality of adolescent relationships with friends and siblings.

Using the Parent Attachment Questionnaire to assess adolescents’ attachment to each of their parents, Boling et al. [38] found a connection between parent-adolescent attachment and social competence, and again a connection between social competence and friendship quality in adolescence. Similarly, using the Kerns Security Scale (KSS) to measure attachment, Lieberman et al. [39] found that friendship qualities such as help, closeness and security in early adolescence, were significantly related to overall security of attachment to both mothers and father. Using the Mother Father Peers Questionnaire to measure parental acceptance, Seginer et al. found that characteristics in adolescent sibling relationships were similar to characteristics in the adolescents’ relationships with both their mother and father [40]. The findings support attachment theory’s proposal that attachment patterns developed already in childhood function as prototypes for future relationships.

The first aim of the current study was to examine the relationship between parental bonding in childhood and friendships in depressed adolescents, as measured with the self-report questionnaires PBI and Adolescent Relationship Scale (ARS) [41]. Based on previous research it was hypothesised that there would be significant associations between reported high levels of parental care and low levels of parental control and reported importance of friendships. The second aim of the study was to examine the connection between adolescents’ reported parental bonding in childhood and sibling relationships in depressed adolescents. It was hypothesised that there would be significant associations between reported high levels of parental care and low levels of parental control and reported importance of siblings.

## 2. Materials and Methods

### 2.1. Study Design

In the current study, data from The First Experimental Study of Transference Work-In Teenagers (FEST-IT) was used [42]. FEST-IT is a randomised, controlled study of psychodynamic psychotherapy for adolescents with MDD, and a collaborative study between the Institute of Clinical Medicine at the University of Oslo and the Clinic for Mental Health and Addiction at Vestfold Hospital. Ethical approval for the current study was obtained from the Regional Committee for Medical and Health Research Ethics (REC).

The study design is further elaborated by Ulberg and colleagues [43].

### 2.2. Patients

The patients were the 70 adolescents included in FEST-IT. One patient withdrew consent, and one patient did not fill in the PBI and ARS self-report forms relevant for the current study. Hence, there were 68 patients, of which 57 were female and 11 were male. They attended lower- or upper secondary school and were aged 16–18 years. The patients were recruited through their referral to the Child and Adolescent Outpatient Clinics in the South-Eastern Health Region, Norway, which represents both urban and rural areas.

Adolescents with a current unipolar MDD diagnosed according to the Diagnostic and Statistical Manual of Mental Disorders, Fourth Edition [10] were included, whereas patients with generalised learning difficulties, a pervasive developmental disorder, psychosis or substance abuse were excluded. Patients with a Beck Depression Inventory-II (BDI-II)-score [44] above 10 and/or a Montgomery and Åsberg Depression Rating Scale (MADRS)-score [45] above 15 were selected. All patients signed an informed consent. Pre-treatment diagnoses that were assessed at baseline are summarised in Table 1. Of the sample, 38.5% of the adolescents lived with both parents, 56.4% lived with one of the parents or commuted between two parents, while 5.1% lived in another housing situation [43].

### 2.3. Instruments

#### 2.3.1. The Parental Bonding Instrument

The Parental Bonding Instrument (PBI) is a psychometric retrospective self-report measure used to assess the experienced parental contribution to the parent-child bond from the child’s point of view (see Appendix B). It asks respondents to recall the level of care and protection they experienced from their parents during the first 16 years of their lives. Each parent is rated separately. It consists of 25 items, where each statement is scaled with a 4-point Likert format ranging from “very like” to “very unlike” the parent that is being rated. 12 items assess the degree of care, i.e., the degree of affection, warmth and closeness shown by the parent, and 13 items assess the degree of protection, i.e., the degree to which the parent allows autonomy and independence. Two scores are obtained for each parent: one care and one control score.

In the present study, a Norwegian-validated PBI was used. Parker et al.’s [23] scoring manual was used to assess the degree of parental bonding(see Appendix A). The Norwegian-validated PBI was incompatible with the scoring manual on two points. Hence, adjustments to the Norwegian version were made to fit the scoring manual.

Firstly, the numbers assigned to the 4-point Likert format in Parker et al.’s scoring manual are Very like = 3, Moderately like = 2, Moderately unlike = 1, Very unlike = 0. On the other hand, in the Norwegian-validated PBI they are Very like = 1, Moderately like = 2, Moderately unlike = 3, Very unlike = 4. Therefore, the Likert format was changed from 1–4 to 3–0 in order for the data to fit Parker et al.’s scoring instructions.

Secondly, in Parker’s et al.’s original PBI items 1, 5, 6, 8, 9, 10, 11, 12, 13, 17, 19, 20 and 23 are positive statements and items 2, 3, 4, 7, 14, 15, 16, 18, 21, 22, 24, 25 are negative statements. In the Norwegian-validated PBI, however, items 1, 2, 5, 6, 9, 10, 11, 12, 13, 14, 17, 18, 19, 20, 23, 24 are positive statements and items 3, 4, 7, 15, 16, 21, 22, 24, 25 are negative statements. For example, in Parker et al.’s original PBI item 2 is worded: “(my mother/father) did not help me as much as I needed”. In the Norwegian-validated PBI the same item is worded: “(my mother/father) helped me as much as I needed”. Therefore, when scoring the Norwegian-validated PBI, items 3, 4, 7, 15, 16, 21, 22, 24, 25 were inverted, rather than 2, 3, 4, 7, 14, 15, 16, 18, 21, 22, 24, 25 as is instructed in Parker et al.’s scoring manual.

#### 2.3.2. The Adolescent Relationships Scale

The Adolescent Relationship Scale (ARS) is a visual analogue self-report scale from 0–10 that assesses the mutual quality of relationships [41]. It consists of nine statements, of which eight statements are presented in pairs and assess the importance of the patient’s relationship with their mother, father, siblings and friends. The patients assess both their own view on the importance of the relationship (i.e., “How much do your siblings mean to you?”) and their view on the relationship’s reciprocity (i.e., “How much do you mean to your siblings?”). The ninth statement assesses the participant’s perceived quality of life (“What is your quality of life at the moment”).

The first four patients included in the study filled in a version of the ARS that assessed the importance of parents as a unit, rather than the importance of mother and father separately. These were discarded and there are therefore 64 patients, rather than 68, that filled in the four statements assessing their relationships to their mother and father. Furthermore, seven of the patients reported having no siblings, and the assessment of importance of siblings therefore had 61 patients.

### 2.4. Procedure

At the first encounter, the patients filled in a BDI self-report form and the therapist assessed the patient using the MADRS. Patients with a BDI-score above 10 and/or a MADRS-score above 15 were selected. The selected patients were then informed of the study and were asked to give written informed consent. After consent was given, the participants partook in a pre-treatment evaluation (baseline) with an external evaluator blinded for randomisation, where they were diagnostically interviewed using the Mini International Neuropsychiatric Interview (M.I.N.I.) 6.0.0 [46], an interview tool used to diagnose psychiatric symptoms according to DSM-IV, and the Structured Interview for DSM-IV Personality (SIDP-IV) [47]. Patients that met the criteria for MDD on the M.I.N.I. were selected and later randomised. 

In addition to the M.I.N.I. and the SIDP-IV, the patients during the pre-treatment evaluation were assessed through a number of measures, therapist questionnaires and self-report questionnaires, such as PBI and ARS. The patients were asked to complete the questionnaires individually in the presence of an evaluator.

### 2.5. Statistical Analysis

The software in the study was SPSS 18. Variables from the PBI were maternal and paternal care, and maternal and paternal control. The mean values of maternal and paternal care and control were computed. High care and control levels were determined according to cut-off points established by Parker, et al. [23]. Scores above 27/24 indicate high levels of maternal/paternal care, and scores above 13.5/12.5 indicate high maternal/paternal control. The total number of individuals with the combinations high care/low control, low care/low control, high care/high control and low care/high control were counted and subsequently placed in the categories “Optimal Bonding”, “Absent Bonding”, “Affectionate Constraint” and “Affectionless Control”. Variables from the ARS were individual items such as “What is your quality of life like now”. The mean value of each item was analysed. A Pearson’s correlation was run to assess the relationship between the variables from the PBI and the items of ARS. The correlation between each variable was analysed. To explore the risk of Type I error, a Bootstrapping analysis was used.

## 3. Results

The mean PBI scores are shown in Table 2. The mean scores of both maternal and paternal care/control are below/above cut-off levels, indicating low levels of care and high levels of control in both mothers and fathers. Mothers are perceived as both more caring and more controlling than fathers.

Based on their maternal and paternal care and control scores, patients were assigned to one of four PBI quadrants. Table 3 presents PBI score distributions for the four types of parental bonding. The distribution between affectionate constraint and affectionless control is similar for paternal and maternal bonding. However, more patients reported optimal bonding with their mother, and more participants report absent bonding with their father.

The mean ARS scores are shown in Table 4. There is an even distribution in the importance of siblings and parents among the participants. However, the statement regarding quality of life and ‘how much you mean to your friends’ have lower means.

In Table 5, the correlations between ARS scores and parental bonding are presented. It was hypothesised that there would be significant associations between reported high levels of parental care and low levels of parental control and reported importance of friendships. There is a statistically significant negative correlation between maternal control and reported importance of friends.

It was also hypothesised that there would be significant associations between reported high levels of parental care and low levels of parental control and reported importance of siblings. There is a statistical significant positive correlation between paternal care and reported importance of siblings.

Moreover, there is a statistically significant positive correlation between maternal care and reported importance of both mother and father, and between paternal care and reported importance of both mother and father. There is a statically significant negative correlation between maternal control and reported meaning of mother. Furthermore, there is a statistically significant negative correlation between maternal bonding and reported quality of life.

Bootstrapping analysis of the significant correlations reported in Table 5 showed similar results. Friends mean to you × Maternal Control of −0.352, (95% CI: −0.558, −0.136; *p* < 0.00), You mean to friends × Maternal Control of −0.269 (95% CI: −0.468, −0.007; *p* < 0.03), Siblings mean to you × Paternal Care of 0.278 (95% CI: −0.41, 0.524; *p* < 0.00), You mean to siblings × Paternal Care of 0.314 (95% CI: 0.42, 0.541; *p* < 0.01).

## 4. Discussion

The results from this study add to a body of evidence linking parenting styles with adolescent psychopathology [25,26,27,28,29,30,31,32,33] and relationships [38,39,40]. The main aim was to determine the relationship between parenting styles, characterised as warmth and control, and the reported importance of relationships with friends and siblings. 

The results partially supported the hypotheses that high levels of parental care and low levels of parental control would be significantly correlated with meaningful relationships with friends and siblings. The findings are consistent with previous research and attachment theory’s argument that attachment patterns developed in childhood are internalised and function as prototypes in future relationships. 

Interestingly, the results showed different relationship patterns for mothers and fathers. High levels of maternal control were significantly correlated with low quality adolescent friendships, whereas high levels of paternal care were significantly correlated to adolescent sibling relationships. Hence, the findings may offer new insight into different effects of maternal and paternal care and control on adolescent relationships.

The results linking high levels of maternal control to low quality of adolescent friendships support existing evidence connecting high levels of maternal control to less positive attachment styles in adulthood [48]. The results also mirror previous findings arguing low fostering of autonomy from mother, but not father, is associated with negative friendship features [38,39].

Research suggests that maternal control has greater negative effects on development than paternal control, and is less tolerated than the latter [49,50,51]. Indeed, the current findings show a significant correlation between maternal control and low levels of both quality of life and reported reciprocity of maternal relationships, whereas no such correlation is found with paternal control. However, limitations on the PBI’s measurement of maternal control should be acknowledged. It has been suggested that high levels of maternal care may be able to temper the negative effects of maternal control, and that the quality of control can vary from intrusiveness to consistent boundary setting [31]. These important nuances may be difficult to differentiate between when using the PBI.

There are two well-established but conflicting theories on how parental factors effect sibling relations: the congruence hypothesis and the compensations hypothesis [52,53,54]. According to the congruence hypothesis, behaviours in the parent-child relationship tend to “spill over” to the sibling relationship [52]. The compensation hypothesis, on the other hand, contends that negative relationships with some family members may be compensated for by more positive relationships with other family members [54,55]. The current study supports the congruence hypothesis, and mirrors previous findings connecting secure attachment styles between adolescent and parent with increased warmth and intimacy in sibling relationships [38,39,40].

The results on sibling relationships and paternal care are particularly interesting in light of a longitudinal observational study by Volling and Belsky [56], that showed connections between facilitating/affectionate fathering and prosocial interactions between six-year-old siblings. This connection was not found between facilitating/affectionate mothering and prosocial interactions between six-year-old siblings. Although Volling and Belsky found a connection between fathering and sibling relationships in children, this connection has, to the best of our knowledge, not been found between fathering and sibling relationships in adolescents. Hence, the current study offers novel findings.

As the patients in the study were all diagnosed with a MDD, attention should be paid to the PBI-score means. The means of parental care and control were below and above cut-off values respectively. The cut-off scores applied in the current study were computed on another population and it is unsure whether they fit the crosslines of high/low care and control in a Norwegian population. Albeit this limitation, the findings may suggest generally low levels of parental care and high levels of parental control in adolescents with depression, and support attachment theory’s argument that parenting style in childhood shapes an internal working model of self that affects one’s disposition to psychopathology, such as depression, later in life [25,26,27,28].

Furthermore, when looking at the relationship between the items of the ARS and the PBI, a correlation was found between the adolescents’ reported quality of life and maternal parenting style. This finding suggests that high levels of maternal care and low levels of maternal control are related to adolescent’s wellbeing. However, no correlation was found between reported quality of life and paternal parenting style. These findings support theoretical [13] and empirically grounded [57] arguments stating that maternal parenting style has a greater effect on the child’s well-being than paternal parenting style. This argument is disputed, as fathers are underrepresented in research on parenting and the parent-infant relationship [58].

Focus must also be given to the means of the ARS. The ARS scores of the patients in the current study showed similarity to the ARS scores of non-depressed adolescents, as found by Hersoug and Ulberg [41]. This finding contrasts with literature stating that depression may have a negative impact on adolescents’ interpersonal relationships [9]. In the current study, two of the ARS-items had seemingly lower scores than the findings made by Hersoug and Ulberg. Firstly, the mean score of the item assessing the patient’s quality of life was lower than in non-depressed adolescents. Secondly, the mean score of the item assessing the reciprocity of the patients’ friendships was lower than in non-depressed adolescents.

Based on the theory of internal working models of self and other, Bartholomew and Horowitz [14] developed an attachment model in adults differentiating between a positive or negative view of self and a positive or negative view of other. Their dichotomisation of the self and other led them to propose four attachment styles: ‘Secure’, ‘Dismissive’, ‘Preoccupied’ and ‘Fearful’. 

In theoretical terms, the ARS can be viewed through the lens of the internal working model. The item measuring “how much your friendships mean to you” can be understood as an indication of the individual’s working model of self, and the item measuring “how much you mean to your friends” can be understood as an indication of the individual’s working model of other. Bartholomew and Horowitz contended that a person with a negative working model of self, but positive working model of other, had a ‘Preoccupied’ attachment style, which could indicate that individual not feeling worthy of love, and striving for validation from their friends in order to feel a sense of self-worth. The findings of the current study could suggest that adolescents with depression have a ‘Preoccupied’ attachment style with friends.

It is important to note that the current study offers a take on correlation, not causality, and the directionality of the influence cannot be determined. Previous research has found bidirectionality in adolescent relationships with parents, friends and siblings. Among notable findings, De Goede et al. [59] found that in addition to parental behaviour shaping adolescents’ perceptions of their friendships, adolescent friendships shaped adolescents’ perceptions of their parents. Furthermore, a study conducted by Rubin et al. [60] suggested that close friendships in adolescence could have a moderating effect on relationships with parents, and findings by Derkman et al. [61] suggest that adolescents may subconsciously perceive their parents as less supportive if the sibling relationships quality drops.

Previous literature has noted that a large range of variation in methodology makes the determinants of attachment unclear. According to Manassis et al. [62], the interchangeable use of the terms ‘bonding’ and ‘attachment’ result in confusion. When testing the construct validity of the Adult Attachment Interview (AAI), often referred to as the ‘gold standard’ of attachment, and the PBI, Manassis found that attachment information obtained from the two instruments were only comparable in participants with optimal attachment histories, but not in participants showing idealisation or anger towards their mother. Based on their findings, Manassis et al. advised caution when using the PBI to obtain information in clinical samples where suboptimal attachment histories are likely, such as the clinical sample in the current study.

### Limitations of the Study

The limitations of the study are primarily the small sample size, the use of subjective measures to assess the transmission of parental bonding and the quality of relationships with others. The PBI has been used with a wide variety of populations, and several studies have demonstrated its strong and stable psychometric properties [63,64,65]. However, due to developmental changes in adolescents’ perceptions of the parent-child relationship questions about the validity of the PBI in adolescents have been raised [66]. As adolescents strive for autonomy, it is possible that they score overprotection more highly than during other developmental periods.

In two of the Bootstrapping analyses, the confidence intervals are mainly positive and negative, respectively, and slightly cross 0. The findings therefore risk a Type I error.

The data were not examined on the basis of the adolescents’ gender, and the female prevalence in the study was 84%.

The ARS utilised in the current study does not specify the qualities of friendship such as companionship, help, closeness and conflict. The word “friends” was not defined for the participants, and could be confused with peer relationships. Previous research has shown that the quality of friendships and peer relations differ in adolescent relationships. Future research could avoid the possibility of misinterpretations by measuring different aspects of friendship or limiting the questions on friendship to encompass the participant’s closest friend.

In future studies the adolescents’ housing situations should be considered. Parental elements, such as age, couple composition, and the presence of psychopathology should be taken into account in order to assess the influence of these variables.

## 5. Conclusions

Through the lens of attachment theory, the current study aimed to further the understanding of attachment implications through a life-span, and specifically the connection between parental characteristics in childhood and relationships in adolescents with depression. Results revealed a statistically significant correlation between high levels of maternal control and low importance of friendship, and a statistically significant correlation between high levels of paternal care and high importance of relationships with siblings. The findings are in line with attachment theory’s postulation that the early child-parent bond becomes internalised and creates a prototype for interpersonal relationships later in life. The results also mirror findings using measurements other than the PBI to assess the connection between the child-parent relationship and adolescent relationships with friends and siblings.

Interestingly, the findings illuminate a difference regarding paternal and maternal parenting styles and their connection to the adolescents’ relationships with friends and siblings. Maternal parenting style was linked to the adolescents’ relationships with friends, whereas paternal parenting style was linked to adolescents’ relationships with their siblings. To the best of our knowledge, this pattern between mother and friends and father and siblings has not been found in previous research, and should be investigated further.

## Figures and Tables

**Table 1 ijerph-19-06530-t001:** The patients’ age, pre-treatment Beck Depression Inventory and Montgomery and Åsberg Depression Rating Scale scores. Diagnoses according to Mini International Neuropsychiatric Interview and Structured Interview for DSM-IV Personality.

	Total (*n* = 68) Mean (SD)
**Age**	17.3 (0.7)
**Beck Depression Inventory**	28.6 (9.1)
**Montgomery and Åsberg Depression Scale**	22.2 (5.5)
	** *N* ** **(%)**
**Female**	57 (84)
**Diagnoses**	68 (100)
Depressive disorder	68 (100)
Social Phobia	19 (28)
Panic Disorder	13 (19)
General Anxiety	17 (25)
Eating disorder	2 (3)
PTSD	2 (3)
**Personality Disorders**	30 (44)
Depressive	24 (35)
Avoidant	19 (28)
Negativistic	3 (4)
Obsessive compulsive	3 (4)
Paranoid	3 (4)
Dependent	2 (3)
Borderline	1 (1)
Histrionic	1 (1)
Schizoid	1 (1)
**More than one Personality Disorder**	17 (25)

**Table 2 ijerph-19-06530-t002:** Parental Bonding Instrument mean scores (standard deviations) in the First Experimental Study of Transference Interference-In Teenagers (FEST-IT).

	Care Mean (SD)	Control Mean (SD)
Maternal	26.2 (7)	15.6 (7.5)
Paternal	22.7 (7.9)	12.8 (5.8)

**Table 3 ijerph-19-06530-t003:** Parental Bonding Instrument score distribution as percentage for the four types of parental bonding in the First Experimental Study of Transference Interference-In Teenagers (FEST-IT).

	Optimal Bonding *N* (%)	Absent Bonding *N* (%)	Affectionate Constraint *N* (%)	Affectionless Control *N* (%)
Maternal	28 (41)	6 (9)	14 (21)	20 (29)
Paternal	21 (31)	14 (21)	13 (19)	20 (29)

**Table 4 ijerph-19-06530-t004:** Adolescent Relationships Scale (ARS) mean scores and standard deviations in the First Experimental Study of Transference Interference-In Teenagers (FEST-IT).

	*N*	Mean (SD)
What is your quality of life like now?	68	4.5 (1.6)
**The importance of friends**		
How much do your friends mean to you?	68	8.5 (1.5)
How much do you mean to your friends?	68	6.8 (2.1)
**The importance of siblings**		
How much do your siblings mean to you?	61	8.7 (1.8)
How much do you mean to your siblings?	61	8.0 (2.1)
**The importance of mother**		
How much does your mother mean to you?	64	8.7 (1.8)
How much do you mean to your mother?	64	9.1 (6)
**The importance of father**		
How much does your father mean to you?	64	8.4 (2.1)
How much do you mean to your father?	64	8.4 (2.2)

**Table 5 ijerph-19-06530-t005:** Correlation coefficients (Pearson r) between Adolescent Relationship Scale and maternal and paternal care and control in the First Experimental Study of Transference Interference-In Teenagers (FEST-IT).

	Maternal Care	Maternal Control	Paternal Care	Paternal Control
What is your quality of life like now?	0.308 *	−0.266 *	−0.041	0.035
How much do your friends mean to you?	0.145	−0.352 **	0.089	0.080
How much do you mean to your friends?	0.196	−0.269 *	−0.034	0.073
How much do your siblings mean to you?	−0.067	−0.139	0.278 *	−0.183
How much do you mean to your siblings?	0.175	−0.157	0.314 *	−0.127

* = Statistically significant at *p* < 0.05 level (2-tailed). ** = Statistically significant at *p* < 0.001 level (2-tailed).

## Data Availability

The data is available from the second author.

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
