# Peer review of "Parental Bonding and Relationships with Friends and Siblings in Adolescents with Depression"

_ijerph, 2022, doi:10.3390/ijerph19116530_

Round 1

Reviewer 1 Report

Thank you for the opportunity to review the paper “A Correlational Study Between Parental Bonding in Childhood and Relationships with Friends and Siblings in Adolescence”. I believe that the study and the topic are relevant to the readers of the International Journal of Environmental Research and Public Health, however major revision is needed. My main concern is related to theoretical background of the presented study and discussion of the obtained results.

Title and Introduction: The title and the theoretical background is not consistent with the findings. The sample is consisted of adolescents with depression diagnosed. There is many scientific work that emphasized the specific functioning of family among depressed adolescents, thus it should be revised in the paper. In Introduction the specific patterns of relationship with parents and other significant others of depressed adolescent should be added, and/or how there are different from the mainstream sample of adolescent (i.e., non-depressed).

Therefore, the theoretical background should be more clearly presented, firstly the mechanism of association between parent-child bond and adolescent relationships with friends and adolescents should be described in more details, especially in the adolescence depression context. After the strong theoretical introduction, Authors could provide information about measurement tools, but firstly the theory should be highlighted, not the questionnaires.

Discussion: Based on the analyses the current discussion is inappropriate. Again, the discussion should be modified and highlight that this associations are investigated among depressed adolescents.

Author Response

Thank you for your comments. Major revisions to the manuscript have been made. The author has reviewed all the points evidenced by the reviewers. Please see the attached Cover Letter.

Kind regards, 

Sarah Fahs

Reviewer 2 Report

SEE ATTACHED FILE, PLEASE.

Author Response

Thank you for your comments. Major revisions to the manuscript have been made. The author has reviewed all the points evidenced by the reviewers. Please view attached Cover Letter.

Kind regards, 

Sarah Fahs

Reviewer 3 Report

In my opinion the article approaches an interesting aspect of attachment implications through life-span, namely on adolescence. This theme is relevant, and the sample studied is very specific; however the article do not consider this added value, and that there are some important aspects that must be improved:

  • the sample of the study is a clinical sample, selected using clinical assessment tools (depressed adolescents). However, this aspect is not considered in introduction. Authors present the study as it won't be focused on a clinical sample. I think that in introduction should be included  a revision about the relationship between attachment and psychopathology, and the hypotheses should be reviewed and discussed from this perspective;
  • as DSM-5 is in use since 2013, authors should update the description of the sample according to actual criteria, and not use Axis I or II as it is not in use in present;
  • Authors should  explain the reason why they decided to use instruments associated to DSM-IV, and the present relevance of results obtained through them;
  • The text in results' section is very repetitive, simply listing the significative correlations founded. Results should be presented in relation with the hypotheses;
  • Authors do not consider the specific sample'characteristics in the discussion and conclusions; in my opinion the discussion should integrate this aspect;
  • In conclusion authors recommend that future studies should distinguish the specificities of mother and father bonding or parental styles; it also should be added the need to be considered the features of parenting : both parents were present in child's life? Mother, father and child lived together or did it ocurred separation or divorce and when? It is important to frame the conclusions, not focusing only the gender of parents.
  • The references must be updated: none of the references is published after 2016. More recent references must be included.

Author Response

(The authors gave the same response as above.)

Reviewer 4 Report

This paper reports a simple correlational study on the associations between parental bonding and items from a scale (ARS) measuring relational aspects. There are several shortcomings of the research, apart from its simplistic design, such as its low sample (64 participants in the sample included in the actual analyses), which I describe in what follows. There are important aspects (commented hereinafter) that I find unclear and that make me unable to appreciate the actual contribution of the study.

line 81 “few studies have used the PBI in order to investigate the connection between parenting behavior as reported by adolescents and adolescents’ ongoing relationships with friends and siblings” – this indicates that there are already studies on the same specific topic. These studies should be presented and briefly reviewed, especially since the research reported here aims to provide a replication of the past studies. Moreover, the exact type of replication that is implemented here should be specified and explained in relation to these past investigations. In this current version, without this information, the contribution of the paper to the extant knowledge in the area cannot be appreciated.

The participants included in this study were patients with depression, which implies a specific population in terms of a clinical feature that is important for both of the major coordinates of the research, i.e., parental bonding and relationships with friends and siblings in adolescence. These relationships should be discussed in the Introduction, and also taken into account in the Discussion.

line 140 “The Norwegian-validated PBI is incompatible with the scoring manual on two points. Hence, adjustments to the Norwegian version were made to fit the scoring manual.” – these differences and adjustments should be described.

line 176 “High care and control levels were determined according to cutoff points established by Parker, et al. [6].” – these cut-off scores were computed on another population than that of the study (i.e., Norwegian), thus it’s unsure whether they fit the crosslines of high/low care and control in thus population. Consequently, the results related to the classification of participants in the four categories should be eliminated (i.e., Table 3 and the paragraph above).

The correlations in Table 5 should be recomputed by eliminating the risk of committing a type I error (i.e., of affirming a significant association that in fact emerged through chance alone, when performing a large set of similar analyses, in this case Pearson correlations). The Bonferonni correction, for instance, could be applied.

Minor issues

The keywords repeat the same concept - Parental Bonding Instrument; should be kept only once; perhaps one could be replaced with the concept of parental bonding;

line 55 there is a reference in APA style that should be eliminated (Daniel, Wassell, & Gilligan, 1999)

section 2.3.2. the response scale of the ARS should be presented.

Author Response

(The authors gave the same response as above.)

Round 2

Reviewer 1 Report

The revised paper addressed my concerns adequately.

Reviewer 2 Report

The authors have made a great effort to improve the scientific quality of this ms. Therefore, I believe that this ms. could be published in this Journal.

Reviewer 4 Report

The authors addressed my previous comments.